# Changes in TRPV1 Expression as Well as Substance P and Vasoactive Intestinal Peptide Levels Are Associated with Recurrence of Pterygium

**DOI:** 10.3390/ijms232415692

**Published:** 2022-12-10

**Authors:** Hatice Deniz İlhan, Betül Ünal, Yusuf Ayaz, Nuray Erin

**Affiliations:** 1Department of Ophthalmology, Faculty of Medicine, Akdeniz University, Antalya 07070, Turkey; 2Department of Pathology, Faculty of Medicine, Akdeniz University, Antalya 07070, Turkey; 3Department of Medical Pharmacology, Faculty of Medicine, Akdeniz University, Antalya 07070, Turkey

**Keywords:** pterygium, recurrence, TRPV1, epithelium, stroma, nerve fibers, substance P, vasoactive intestinal peptide

## Abstract

Pterygium, a disease of the ocular surface, is characterized by the proliferation and invasion of fibrovascular tissue. Chronic inflammation contributes to pterygium occurrence. Sensory neuropeptides of TRPV1-positive nerve fibers are involved in inflammation and corneal wound healing. The possible association between TRPV1 in nerve fibers and neuropeptides such as Substance P (SP) and Vasoactive Intestinal Peptide (VIP) in the recurrence of pterygium has not been examined before. The pterygia from 64 patients were used to determine changes in SP and VIP levels using 10 min acetic-acid extraction that yielded mainly neuronal peptides. There was a sufficient amount of pterygium tissues from the 35 patients for further immunohistochemical analysis of TRPV1 and S100, which is a glial marker to visualize nerve fibers. SP and VIP levels increased markedly in cases with primary and secondary recurrences, and there was a close correlation between SP and VIP levels. TRPV1 expression increased in the epithelium, while stromal expression decreased in recurrences. Nerve fibers were demonstrated mainly in the stroma, and serial sections confirmed the localization of TRPV1 with the nerve fibers. These results together with previous findings demonstrated that the increased epithelial expression of TRPV1 in recurrent pterygia might be involved in the pathogenesis, and the inhibition of epithelial TRPV1 activity may prevent recurrence.

## 1. Introduction

Pterygium is a disease of the ocular surface that is characterized by the proliferation and invasion of fibrovascular tissue from the bulbar conjunctiva to the cornea. Chronic inflammation may contribute to pterygium occurrence. It was postulated that inflammation at the junction of the conjunctival blood vessels and Bowman’s membrane, which attracts conjunctival blood vessels on the cornea, may cause pterygium [1]. Specifically, inflammation induced by ultraviolet B (UV-B) radiation may contribute to the pathogenesis of pterygium [2]. In accordance, levels of inflammatory cytokines such as IL-6, IL-8 and tumor necrosis factor-α (TNF- α) are increased in pterygium [3,4,5].

Surgical removal is necessary for the treatment of pterygium in the event of ocular surface changes. Recurrence is the main postoperative problem, which can be seen in up to 80% of patients [6]. Prediction of the factors affecting recurrence is not fully understood. Pterygium follows the radial extensions of the corneal nerves; hence, nerve fibers may play an important role in its development and, perhaps, in recurrence [7,8,9]. However, to our knowledge, innervation of the pterygium has not been studied before.

TRPV1, a receptor for capsaicin, is found in peptidergic sensory nerve fibers. TRPV1-positive sensory nerve fibers are involved in local inflammatory response via the release of neuropeptides, such as Substance P (SP) [10]. Sensory neuropeptides of TRPV1-positive nerve fibers play a role in inflammation and corneal wound healing [11,12,13,14]. Specifically, SP is a potent chemoattractant for fibroblasts and vascular endothelial cells, such that SP might be involved in the profibrogenic and angiogenic mechanisms of pterygium tissue [9]. However, only a few studies have examined the possible changes in SP levels in pterygium in relation to recurrence [15].

Vasoactive intestinal peptide (VIP) is another neuropeptide that may modulate inflammatory responses in the eye. VIP is found in the glands around the eyelid and ocular surface [16]. In addition, VIP accelerates endothelial wound closure of the cornea [17]. VIP is co-localized with SP in rat episclera [18], and VIP-immunoreactive nerve fibers are in close contact with substance P immunoreactive fibers in the skin of human faces [19].

TRPV1 expression has been shown in pterygium tissue [20]. However, the possible role of TRPV1, nerve fibers and neuropeptides in the recurrence of pterygium has not been examined before. Hence, in the present study, we aimed to examine the possible relationship between SP, VIP and TRPV1 in cases with a recurrence of pterygium.

## 2. Results

The study included 39 males and 25 females with average ages of 51.5 (min: 27, max: 71) and 57.6 (min: 31, max: 75). Eleven of forty-four primary cases experienced recurrence within 24 months (3–24 months). Pterygium recurred in six of the relapse cases. All of those who had recurrent surgery had at least one and at most six pterygium surgeries. The minimum follow-up period for patients was 24 months. The cases included in the study were divided into the following three groups: a primary group (group 1), recurrence of primary group (group 2) and a group that had recurrence surgery (group 3). The demographic features of these groups can be seen in Table 1.

### 2.1. Changes in SP and VIP Levels

As seen in Table 2, the SP and VIP levels of pterygium tissue in the primary group who did not have a recurrence in the follow-up period (group 1) were markedly lower compared to the primary pterygia of patients who had recurrence (group 2). Similarly, SP as well as VIP levels of patients with multiple recurrences (at least one recurrence at the time of surgery, group 3) were markedly higher compared to group 1. We also analyzed the data based on the number of recurrences and the time between the last two recurrences (Table 3). Patients who had recurrence within 6 months had significantly higher levels of SP compared to patients with recurrence after 6 months.

A close correlation between SP and VIP levels was observed in all three groups as determined by linear regression. The Pearson correlation coefficient (r) was more than 0.9. Specifically, r-values were 0.97; 0.98; 0.92 for group 1, group 2, and group 3, respectively (Figure 1).

### 2.2. Changes in TRPV1 Expression

Diffuse TRPV1 expression was observed in the pterygium. As seen in Figure 2, both the epithelium and the stroma were positively stained.

### 2.3. TRPV1 Expression in Epithelium and Stroma (Table 4)

Diffuse but low-intensity staining was observed in half of the primary pterygia cases such that more than 50% of cells were TRPV1 positive (9 out of 18 cases, group 1). Staining intensity was high only in three cases (3/18). Epithelial staining was not observed in one case.

**Table 4 ijms-23-15692-t004:** Expression. Distribution and Intensity of TRPV1 and S100 in Group 1, 2, 3. * *p* = 0.035 compared to group 1 epithelium. ** *p* = 0.0095 compared to group 1 stroma. 3–4+ represents distribution area of 50% and more. 2–3+ represents distribution area of 5–15% for S100 staining.

3–4+		TRPV1 Distribution	TRPV1 Intensity	S100 Distribution (2–3+)
Group 1	Epithelium	9/18	3/18	
	Stroma	6/18	10/18 *	12/18 (67%)
Group 2	Epithelium	4/8	1/8	
	Stroma	1/8	0/8 **	5/8 (62.5%)
Group 3	Epithelium	4/9	0/9	
	Stroma	1/9	3/9	4/9 (44%)

In group 2 (primary pterygia which reoccurred within 24 months), TRPV1 were positively stained in the epithelium in all cases (n = 8), and more than 50% of cells were TRPV1 positive in half of the cases (4 of 8 cases).

In group 3 (pterygium of multiple reoccurrences), TRPV1 was positively stained in the epithelium in all cases (n = 9), and more than 50% of cells were TRPV1 positive in four of nine cases. Staining intensity was low to moderate in all groups.

### 2.4. TRPV1 Expression in Stroma (Table 4)

Group 1. Stromal staining was observed in all samples. Over 50 % of cells were TRPV1 positive in six out of eighteen primary pterygia cases. Intense staining (3+) was observed in 10 out of 18 cases. The intensity of stromal staining was significantly higher than that of the epithelium when a staining intensity of 3+ was compared within this group (Fisher’s exact test).

Group 2. Stromal staining was observed in all of the samples except one. Over 50% of cells were TRPV1 positive in one out of eight cases. Intense staining (3+) was not observed in any case, and intensity was mostly moderate. The area of staining was markedly greater in the epithelium compared to the stroma in this group when staining areas of 75–100% (grade 4) were compared (3/8 vs. 0/8) (Fisher’s exact test). (In Table 4, grade 3 and 4 are shown for distribution). Interestingly, the staining intensity of the stroma (3+) was markedly lower in group 2 compared to group 1.

Group 3. Stromal staining was observed in all samples except one. Staining over 50% was found in one out of nine cases. Intense stromal staining (3+) was observed in three out of nine cases. Staining intensity and the area of staining were not significantly different among the epithelium and the stroma in this group.

### 2.5. S100 Immunoreactivity

S100 is the marker for glial cells and was used to visualize nerve fibers (Raponi, E., 2007). Intense punctuate staining was observed in all groups, and the area of S100 staining was similar among groups (Table 4) (Figure 3). The immunoreactivity score for S100 was 1.94+/−0.19, 1.75+/−0.31,1.67+/−0.37 for groups 1, 2 and 3, respectively, demonstrating that nerve fibers constitute, on average, 5–10 % of the stromal area. S100 staining was performed in sections of the tissues to determine possible co-localization of TRPV1 with S100. As seen in Figure 3, TRPV1 immunoreactivity was present in areas of S100 positivity.

## 3. Discussion

We have demonstrated for the first time that both SP and VIP levels increased markedly in cases with primary and secondary recurrences (groups 2 and 3, respectively) and that there was a close correlation between SP and VIP levels. TRPV1 expression increased in the epithelium, while stromal expression decreased in primary recurrences (group 2) compared to non-recurrent cases (group 1). In addition, TRPV1 immunoreactivity was stronger in the stroma compared to the epithelium of non-recurrent cases. Nerve fibers were demonstrated mainly in the stroma, and serial sections confirmed the localization of TRPV1 with the nerve fibers. Changes in the stroma and epithelial expression of TRPV1 in group 3 were somewhat similar to group 2, but the differences were not statistically significant. These variations might be due to the presence of previous surgery for pterygium/pterygia, such that post-surgical changes may have altered the structure of newly formed pterygium. Specifically, it was shown that corneal subbasal nerve fibers deteriorated in pterygium patients, and pterygium excision improved corneal sensitivity and increased corneal subbasal nerve density [21].

TRPV channels that are sensitive to heat are present in conjunctival epithelial and endothelial cells, and TRPV1, TRPV2 and TRPV3 activities are considered essential homeostatic mechanisms for corneal endothelial function under different ambient conditions [22,23]. In human corneal endothelial cells, capsaicin, a selective TRPV1 agonist, induces Ca2+ transients [23]. Furthermore, the exposure of human corneal epithelial cells to hypertonic stress elicits TRPV1 activation, leading to increases in pro-inflammatory cytokine release via mitogen-activated protein kinase and NF-κB signaling [24]. Inflammation is considered an important factor in the occurrence and recurrence of pterygium [3,4,5,25,26]. There are a few reports about the expression of TRPV1 in pterygium. It was shown that there is a significant correlation between TRPV1 and TRPV3 expression in pterygium; this, however, is not associated with recurrence [20]. It was also shown that 90% of TRPV1 immunoreactivity is localized in the epithelial layers of pterygia in a small group of both primer and recurrence cases [20]. Garreis et al. also demonstrated a similar pattern of TRPV1 staining in pterygia as well as expression in human pterygial epithelial cells. In addition, they have documented that TRPV1 may mediate the mitogenic effects of growth factors and suggested that TRPV1 might be a potential therapeutic candidate in pterygium [27]. Our findings are somewhat in agreement with these findings and further document a shift from stromal staining towards the epithelium in recurrent pterygium. Specifically, we observed an increase in TRPV1 in the epithelial layer in cases with recurrence, while the opposite occurred for stromal expression. Our findings, together with previous findings, suggest that inhibition of epithelial TRPV1 may inhibit the growth of pterygium.

TRPV1 is expressed in both neuronal and non-neuronal cells. It was shown that TRPV1+ nerves account for ~40% of nerve fiber length in the intact corneal epithelium and ~80% in the stroma and that following injury, TRPV1+ nerve density increases in a mouse model [28]. The activation of TRPV1+ nerves in the eye induces the release of neuropeptides such as neurokinins, calcitonin gene-related peptide (CGRP), and substance P (SP) [29]. In accordance, we observed a significant innervation of the stroma of pterygium with nerve fibers in all groups. In addition, short acetic acid extraction of neuronal peptides [30] yielded a significant amount of SP and VIP, which was markedly increased in recurrent pterygia. Given the fact that patients in group 2 (primary recurrence) did not have previous surgery, these changes are directly related to the recurrence capacity of the pterygium. It was previously shown that expression of the NK1 receptor, the receptor for SP, is increased in pterygia, and SP induces cell migration in pterygium fibroblasts which is inhibited by an NK(1) receptor antagonist [9]. Hence, it was postulated that SP contributes to the shape of the lesion through its profibrogenic and angiogenic actions. Specifically, NK-1 antagonists reduced both corneal hemangiogenesis and lymphangiogenesis in two mouse models of corneal neovascularization [31]. Our results verify to some extent and further demonstrate that SP might be involved in recurrence. The density of nerve fibers was not markedly different among groups, hence increased neuronal peptide levels are likely to be due to the sensitization of these nerve fibers. Therefore, identification of the factors that lead to these changes is required to find effective preventive measures. Along these lines, TRPV1 modulators might be effective in decreasing neuropeptide contents, consequent inflammation and recurrence.

SP has a dual function such that while it has profibrogenic and angiogenic actions, a lack of SP or inhibition of SP activity may lead to different pathological changes in the eye. SP participates in the early phase of wound healing [32], and a lack of SP promotes neurotrophic keratitis [33,34]. Similarly, SP promotes wound healing in the diabetic cornea and induces nerve fiber regeneration to restore corneal sensitivity. On the other hand, blocking SP activity results in pathological diabetic epithelial changes and reduced corneal sensitivity in mice [35,36]. Hence, the modulation of SP activity rather than complete inhibition might be a better approach for proliferative eye pathologies.

Studies suggest important regulatory roles for VIP in the eye. Specifically, nerve fibers in the human lacrimal gland interstice mainly express VIP [16,37]. VIP+ fibers are also present close to the acini of human meibomian glands [38]. VIP is present along the epithelial–stromal junction and adjacent to goblet cells, and stimulates mucin secretion by conjunctival goblet cells [39]. Similarly, VIP is involved in smooth muscle relaxation and the excretion of secretions in the intestinal system [40]. VIP also has important anti-inflammatory and anti-viral effects [40,41,42]. Hence, the increased levels of VIP observed here might be a protective mechanism for the increased inflammation observed in pterygia.

Our findings demonstrate that VIP levels are increased in recurrent pterygium, and the levels closely correlate with SP levels. Given the fact that TRPV1 expression is co-localized with the majority of stromal nerve fibers and the close correlation among SP and VIP levels, it is likely that both of these peptides are present in TRPV1+ sensory nerve fibers. Similarly, co-localization was demonstrated in rat episclera [18]. VIP-immunoreactive nerve fibers are in close contact with substance P immunoreactive fibers in skin biopsies of the human face [19]. Hence, the inhibition of TRPV1+ sensory nerve fibers may lead to a sustained decrease in VIP, which may have detrimental consequences. Similarly, it was shown that TRPV1 inhibition augments inflammation and N-Oleoyl dopamine, a TRPV1 agonist, has strong anti-inflammatory actions [43,44]. These aspects should be further evaluated given the fact that TRPV1 antagonists are in clinical trials. Specifically, Tivanisiran (SYL1001) is a small interfering RNA (siRNA) targeting TRPV1, administered in solution as eye drops and is currently in phase 3 clinical research for Dry Eye Disease treatment (NCT05310422). Although the short-term use of Tivanisiran was found to be safe, long-term use should be evaluated in more detail.

In conclusion, our results, together with previous findings, suggest that the increased epithelial expression of TRPV1 found in recurrent pterygia is likely to be involved in pathogenesis, and inhibition of epithelial TRPV1 activity may prevent recurrence. On the other hand, complete inhibition of the TRPV1+ stromal nerve fibers in pterygium may hinder the protective effects of neuropeptides such as VIP. Furthermore, SP also has dual functions such that it induces inflammation but also enhances wound healing and is part of a defense system involving corneal protection. Hence, the modulation of SP and neuronal TRPV1 activity rather than complete inhibition seems to be required for the prevention of pterygium as well as for safety.

## 4. Materials and Methods

The local ethics committee approved this randomized prospective study (Approval Number 3005.2018/392). The study was carried out in compliance with ethical standards, and informed consent was obtained from all patients who participated in this study. Pterygium samples taken from patients who were operated on at the Akdeniz University Ophthalmology Department were used. Exclusion criteria were autoimmune diseases affecting the ocular surface (e.g., Sjögren, rheumatoid arthritis), diabetes, uveitis, scleritis, those who had ophthalmic surgery or ocular trauma other than pterygium surgery, facial paralysis, glaucoma, as well as patients receiving steroids and non-steroid anti-inflammatory drugs. The cases were evaluated in terms of the morphology of the pterygium using the grade defined by Tan et al. [25]. All cases were grade 3, whether primary or recurrent, with a vertical length extending more than 3 mm from the limbus to the center of the cornea. Samples of forty-four primary and twenty recurrent cases with at least 24 months of postoperative follow-up were included in the study.

An experienced surgeon (HDI) performed all surgeries, and the procedure was pterygium surgery with conjunctival autograft in all patients. The excised pterygium tissue was divided into two parts. One part was transported in liquid nitrogen to the laboratory and stored at −80 °C. The remaining part was sent for pathological examination in formaldehyde.

### 4.1. Measurement of SP and VIP Levels

Tissues from 64 patients stored at −80 °C were weighed with a precision scale. Each tissue was collected in 1.5 mL tubes, phosphate-buffered saline was added, and the washing procedure was performed with centrifugation at 9000 rpm for 4 min. After PBS was discarded, 2% acetic acid was added, and extractions were obtained by performing incubation at 95 °C for 10 min, as described before [30]. Peptide levels obtained within 10 min of acetic acid incubation were mainly derived from neuronal tissue, as demonstrated before for SP levels [45]. VIP levels were measured using a Human VIP ELISA Kit (Elabscience cat. no.: *E*-EL-H2155, TX, USA), and SP levels were measured using an EIA kit (Cayman, cat. no.:583751, MI, USA) according to the manufacturer’s instructions. From each sample, 25 and 50 μL were used for the immunoassay of peptides, which provided results that were within the confidence interval (95%).

### 4.2. Histopathological Evaluation

There was a sufficient amount of pterygium tissues from the 35 patients for further immunohistochemical (IHC) analysis of TRPV1 and S100. For IHC staining, samples were prepared on a positively charged slide with a tissue thickness of 4 μm. The tissue preparations were partially deparaffinized. In accordance with the protocol, the samples were stained using a mouse monoclonal anti-S100 (Thermo Scientific (Clone 4C4.9; 1:100), Waltham, MA, USA) and a rabbit polyclonal TRPV1 antibody (Novus Biologicals (NBP171774; dilution, 1:100), Littleton, CO, USA). In order to determine possible co-localization, consequent sections were stained with TRPV1 and S100. An experienced pathologist conducted the analysis blindly under a light microscope.

The epithelium, connective tissue, and the area of nerve fibers were evaluated separately for TRPV1 staining. Because TRPV1 positivity was more diffuse, a four-stage immunoreactivity scale was used to evaluate the localization (grade 0, no stained cells; grade 1, <25%; grade 2, 25–50%; grade 3, >50–75%; and grade 4, >75%). Staining intensity was graded as follows: 0, negative; 1, weak; 2, moderate; and 3, strong. S100 staining was used to determine neuronal innervation of the pterygium. Nerve fibers were scarce but were strongly stained; hence, the staining intensity was 3 in all samples. To obtain an immunoreactivity score, the percentage of stained cells in this group was graded differently, as follows: grade 0, no stained cells; grade 1, <5%; grade 2, 5–10%; grade 3, >10–15%. The immune reactivity score was calculated by multiplying the immune reactivity scale by the percentage. A statistical analysis was performed using both intensity and distribution values separately as well as immune reactivity scores.

### 4.3. Statistical Analysis

A statistical analysis was performed using the GraphPad InStat program (CA, USA). Data were analyzed using the chi-square test for categorical variables, or the *t*-test for independent variables. A *p*-value of less than 0.05 was considered statistically significant. Correlation analyses were performed by calculating Pearson’s correlation coefficient.

## Figures and Tables

**Figure 1 ijms-23-15692-f001:**
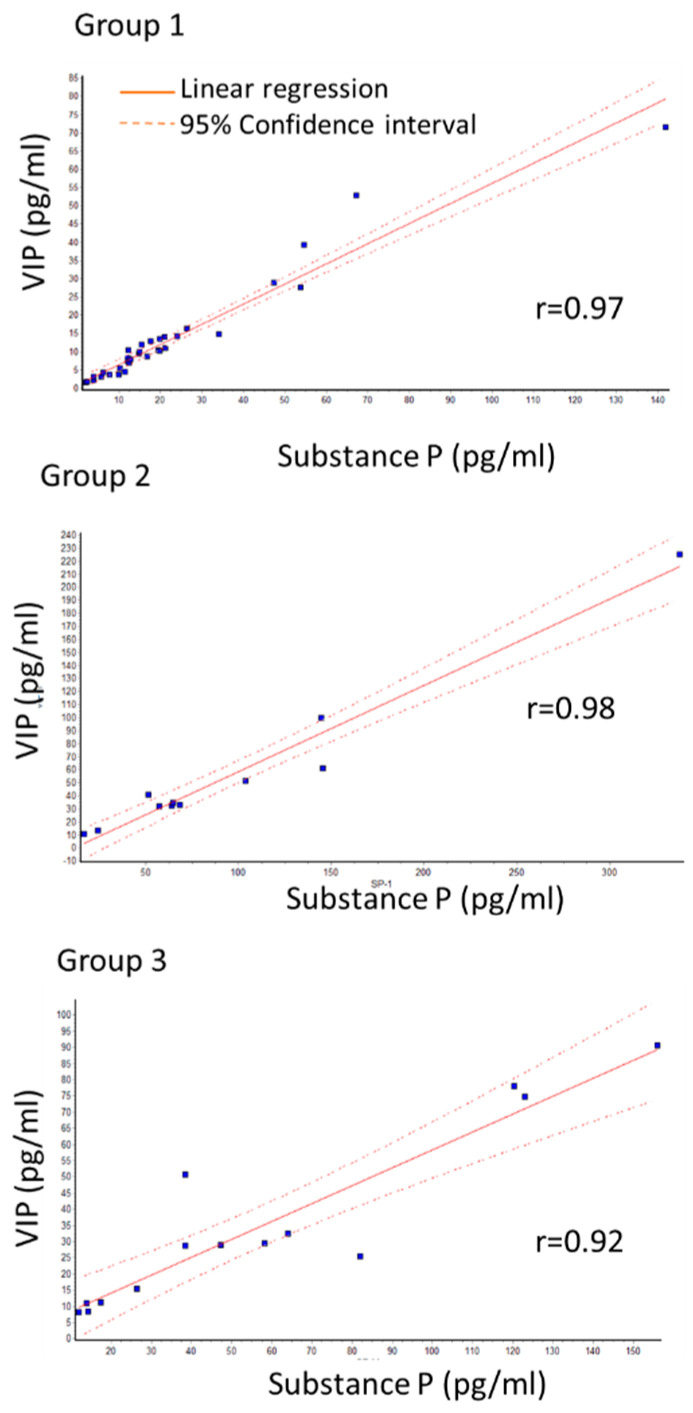
Correlation between SP and VIP levels in primary group (group 1) with no recurrence, recurrence of primary group (group 2) and group that had recurrence surgery (group 3). r shows Pearson correlation coefficient. Close correlation is seen in all groups.

**Figure 2 ijms-23-15692-f002:**
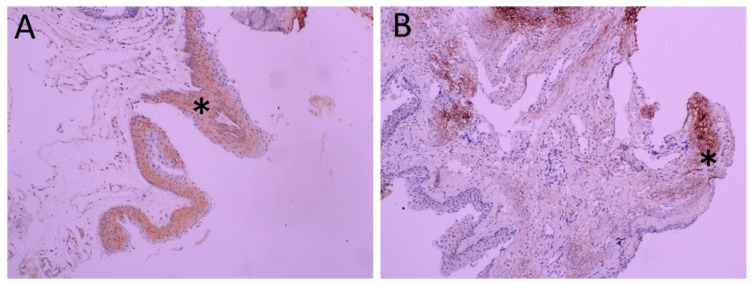
Immunohistochemical analysis of TRPV1 in pterygia. Representative figure of diffuse and intense staining of epithelium is seen in Panel (**A**) (*). Intense staining of stroma (*) is seen in Panel (**B**) at 200 magnification.

**Figure 3 ijms-23-15692-f003:**
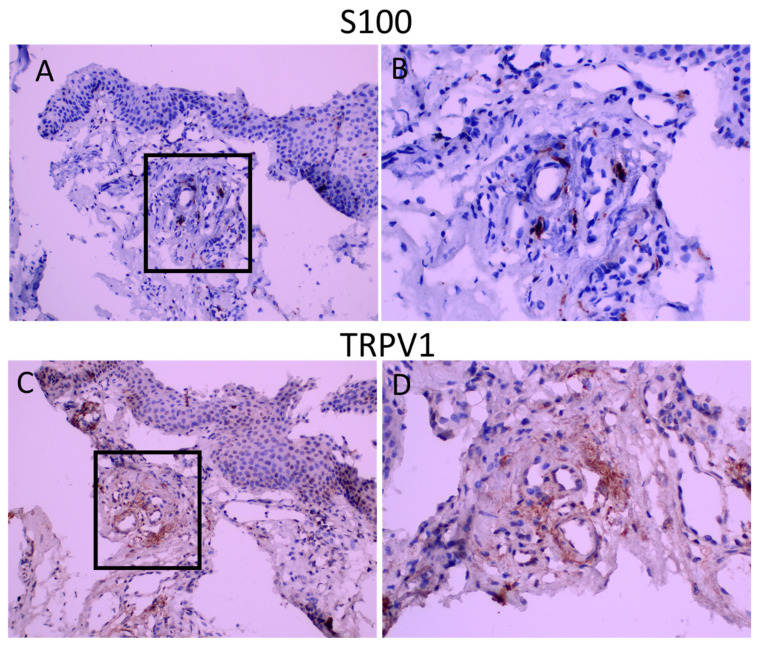
S100 and TRPV1 staining in consequent sections. Panel (**A**,**C**): appearance at 200 magnification. Panel (**B**,**D**): appearance of demarcated areas at 400 magnification. TRPV1 staining is diffuse and covers the areas of S100 immunoreactivity.

**Table 1 ijms-23-15692-t001:** Demographic features. * *p* < 0.05 compared to group 2. ‘n’ denotes number of patients.

	Group 1	Group 2	Group 3
Gender (n) M/F	18/15	7/4	14/6
Age	54.82 ± 5.48	52.36 ± 5.03	53.55 ± 2.99
Pterygium vertical length	3.77 ± 0.11	4.5 ± 0.3	4.02 ± 0.21
Duration between recurrences (month)	0	7.73±1.16	81 ± 17.8 *
Recurrence number	0	1	2.3 ± 0.39

**Table 2 ijms-23-15692-t002:** Levels of SP and VIP.

	Group 1 (n= 33)	Group 2 (n= 11)	Group 1 vs. Group 2	Group 3 (n= 20)	Group 1 vs. Group 3
VIP	13.59 (±2.63)	57.38 (±18.31)	*p* < 0.001	35.60 (±6.4)	*p* < 0.05
SP	22.99 (±4.61)	98.14 (±27.13)	*p* < 0.001	56.94 (±10.18)	*p* < 0.05

**Table 3 ijms-23-15692-t003:** SP and VIP values in relation to time between last two recurrences * *p* = 0.0134 compared to the group that recurred after 6 months. ‘n’ denotes number of patients.

	Recurrence after Last Surgery (n)	SP	VIP
0–6 months	11	112.53 ± 25.63 *****	57.37 ± 24.79
After 6 months	6	43.39 ± 9.75	41.83 ± 11.42

## Data Availability

Not applicable.

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
