# Peer review of "Changes in TRPV1 Expression as Well as Substance P and Vasoactive Intestinal Peptide Levels Are Associated with Recurrence of Pterygium"

_ijms, 2022, doi:10.3390/ijms232415692_

Round 1

Reviewer 1 Report

Congratulations to the authors for a well done study that contributes importantly to the field, especially as TRPV1 antagonist therapeutics become available.

Consider making a matrix/table showing recurrence risk and ICC levels of the TRPV1 location, SP and VIP levels on a relative scale (e.g. + to ++++)

Grammar, plurals, and typos in the discussion section.  minor fixes.

Author Response

Comment: Consider making a matrix/table showing recurrence risk and ICC levels of the TRPV1 location, SP and VIP levels on a relative scale (e.g. + to ++++)

Response: Thank you very much for the suggestion. We found a correlation between recurrence and SP as well as VIP levels that was shown in table 3. We did not find any correlation between ICC results and peptide levels as explained in results section. Hence, to prevent repetition and possible confusion we did not include ICC levels of the TRPV1 expression, recurrence, SP and VIP levels on a relative scale.

For the reviewer, we  included the SP levels in relation to TRPV1 expression in pdf attached.

Manuscript was proofread for minor errors.

Reviewer 2 Report

The recurrent rate of primary pterygium excision in your study was about 33%.

About table 1, recurrence number of group 1 should be 11?

Was there any specific finding of recurrent case in group 2 with longitudinal observation?

Moreover, were there differences in surgical methods between primary and recurrent groups?

Author Response

Comment: The recurrent rate of primary pterygium excision in your study was about 33%. About table 1, recurrence number of group 1 should be 11?

Response: Group 1 includes the patients with primary pterygium whom did not have a recurrence during follow up. Hence recurrence in this group is zero. Recurrence group is group 2 in which n=11.

Comment: Was there any specific finding of recurrent case in group 2 with longitudinal observation?

Response: Because the number of patients were limited in recurrence group we could not detect any specific association with the parameters studied.

Comment: Moreover, were there differences in surgical methods between primary and recurrent groups?

Response: No, the same surgeon performed all operations and the procedure was pterygium surgery with conjunctival autograft in all patients. This information has been included in method section and highlighted in red. .